# Meditation and Five Precepts Mediate the Relationship between Attachment and Resilience

**DOI:** 10.3390/children9030371

**Published:** 2022-03-07

**Authors:** Justin DeMaranville, Tinakon Wongpakaran, Nahathai Wongpakaran, Danny Wedding

**Affiliations:** 1Graduate School, Chiang Mai University, Chiang Mai 50200, Thailand; justinross_de@cmu.ac.th (J.D.); nahathai.wongpakaran@cmu.ac.th (N.W.); danny.wedding@gmail.com (D.W.); 2Department of Psychiatry, Faculty of Medicine, Chiang Mai University, Chiang Mai 50200, Thailand; 3Department of Clinical and Humanistic Psychology, Saybrook University, Pasadena, CA 91103, USA; 4Department of Psychology, University of Missouri-Saint Louis, St. Louis, MO 63121, USA

**Keywords:** attachment, anxiety, avoidance, resilience, meditation, precept, adolescent, boarding school, Buddhism

## Abstract

Secure attachment is fundamental to the development of resilience among adolescents. The present study investigated whether meditation and precept practices influence the relationship between attachment and resilience. This study recruited 453 10th–12th-grade boarding school students who completed the Experience of Close Relationship Questionnaire (revised), Resilience Inventory, Inner Strength-Based Inventory, and Precept Practice to assess attachment, resilience, meditation practice, and precepts adherence. The participants’ mean age was 16.35 ± 0.96 years; 87.9% were females, and 89.2% were Buddhists. A parallel mediation model within the structural equation framework was used for an analysis of the indirect effect of attachment on resilience through meditation and precept practices. The indirect effects of attachment anxiety and avoidance on resilience were β = −0.086, 95% CI = −0.125, −0.054, *p* < 0.001, and β = −0.050, 95% CI = −0.088, −0.021, *p* = 0.006, respectively. The indirect effect size resulting from meditation was significantly higher than that resulting from observance of the precepts. The parallel mediation model explained the 33% variance of the resilience scores, compared with 23% from the direct effect of attachment anxiety and avoidance only. This work provides evidence that meditation and precepts significantly affect the relationship between attachment and resilience.

## 1. Introduction

Attachment refers to an emotional bond between two individuals. It develops initially through psychological connectedness and dependence between an infant and caregiver [1]. Caregiver–infant attachment influences the formation of a child’s internalized strategy for coping with distress that acts as a foundation for interaction with other people [2]. Attachment strategies are categorized into secure and insecure, the latter having two patterns: anxiety (preoccupied) and avoidance (dismissing) [1]. Modern attachment researchers identified fearful (disorganized) as a fourth attachment pattern characterized as a combination of anxious and avoidant types [2,3]. Insecure attachment is a risk factor for psychopathology, and also found more frequently in clinical populations, but it is not in itself a psychopathology [2]. Caregiver–infant attachment has a significant impact through life and relationships, having an enduring influence on future attachment bonds [4,5] and a significant influence on behavior [6] and mental health [7,8,9,10,11]. Whereas an insecure person will struggle with distress and emotional regulation [7], a secure person can draw on an inner sense of capability for coping with distress [12]. A secure attachment style predicts positive attributes that can become a buffer for psychological derailment [13,14]. This positive adaptation amidst adversity is a definition of resilience, a characteristic which has a significant correlation across studies with secure attachment [15,16]. The effect of attachment on resilience is not only viewed as a personality feature or a psychological process, but can also be interpreted as evidence of psychophysiological resilience, as secure attachment predicted higher levels of warmth-liking, physiological quiescence, and fewer negative feelings, even during social exclusion [17]. A study that included both children and adults showed correlations between attachment scores and resilience scores, effective distress management, and better coping responses [18,19].

In Thailand, where 95% of the population identify as Theravada Buddhist, meditation is only one part of a canon of teachings about distress management. A first practice within Theravada Buddhism is following the Five Precepts (Sila): refraining from killing (e.g., insects and all living beings), theft (e.g., taking things that belong to others without permission), sexual misconduct (e.g., adultery), false speech (lying), and the consumption of intoxicants (e.g., alcohol). Precept adherence serves to regulate social behavior; in addition, as a religious practice, the precepts protect the individual and prepare him or her for meditative practice. A connection between parental attachment and Buddhist Thai adolescents was documented in research that found parent intrinsic religiosity to be associated with teen spiritual practices (i.e., meditation and observance of the precepts) [20]. Intrinsic religious orientation, which includes using religious teachings to guide behavior, has been shown to be a protective factor in adolescents against early-onset alcohol use [21] and teenage delinquency [22]. These risk behaviors (non-adherence to precepts) cluster among Thai adolescents [23,24], and individual insecure attachment styles that are associated with lower affect regulation, higher impulsive behaviors [25], and higher incidents of substance use [26], and they may increase other behavioral risks [24].

Within Theravada Buddhism, the precepts are practiced in tandem with meditation [27,28]. Meditation includes a range of methods that are categorized into two types, i.e., samatha (e.g., breathing meditation, mindfulness of the body, recollection of dhamma/Buddhist teaching), and vipassana or insight meditation (e.g., impermanence). There is currently tremendous interest in mindfulness as a tool in psychotherapy; however, the term ‘mindfulness’ is often used synonymously with ‘meditation,’ and different mindfulness practices often overlap and are difficult to distinguish from the methods and goals of different types of meditation [29,30]. Mindfulness research, however, does provide additional insight into the psychological processes underlying meditative experiences.

Amongst adolescents, meditative interventions for distress management were shown to be more effective when compared with adults [31]. Regarding meditation practice, attachment security increased the likelihood that new meditators continued to meditate while also increasing distress regulation [32]. Meditation frequency was also associated with higher emotional intelligence, self-efficacy, and lower perceived stress amongst Thai people [33]. Attachment security has been shown to be associated with mindfulness (awareness, receptiveness), which theoretically supports a coherence of mind that impacts psychological and physical health, emotional regulation, and relationship quality [34]. Attachment anxiety and avoidance were strongly and negatively associated with mindfulness [35,36] and resilience [37]. One mindfulness meditation intervention was shown to increase the resilience of an experimental group compared to a control group [38]. An incarcerated western population decreased their avoidance of thoughts related to alcohol use following a meditative intervention, and demonstrated resilience against later substance use [39]. Regular meditation was also found to be associated with lower perceived stress, an important factor related to resilience [40]. Unlike meditation, no direct evidence regarding the relationship between Buddhist precept adherence and resilience has been reported.

While the relationships between attachment and resilience, attachment and meditation, and meditation and resilience have been studied, little is known about how precept observance may impact resilience. The objectives of the present study were to investigate correlations between attachment dimensions, meditation practice, five precept adherence, and resilience amongst boarding school students. Day school students and boarding school students have similar parental attachments; however, boarding school students live a more structured, homogenous school life with fewer confounding factors [41].

The researchers hypothesized that secure attachment would be positively associated with meditation frequency, adherence to the five precepts, and resilience. Insecure attachment, however, was hypothesized to be negatively associated with meditation frequency, adherence to the five precepts, and resilience. Meditation practices and precept observances were hypothesized to mediate (i.e., reduce the magnitude of the relationship) insecure attachment and resilience. No prior empirical studies were found documenting the relationship between meditation practice and precept practice; therefore, the researchers could not predict direction between mediators; as a result, a parallel mediation model was proposed (Figure 1).

## 2. Materials and Methods

The present study recruited 453 male and female students in 10th–12th grade, aged 15 and older, from Thai boarding schools in Northern Thailand between July and August 2021. The schools were purposively selected considering similar socioeconomic status, the number of students, and the female-to-male ratio. Two Buddhist schools in urban areas of two provinces and three secular schools in urban and suburban areas from three provinces participated. All meditative styles practiced by students were accepted. Students with special needs and students who were blind or deaf were excluded. Informed consent was obtained, and almost all participants received parental permission to participate. For students whose parents were unavailable or difficult to reach, the school administrators’ consent was obtained on behalf of the parents. This study was approved by an ethics committee, Faculty of Medicine, Chiang Mai University.

### 2.1. Instruments

Sociodemographic information about age, sex, and family monthly income was collected. Others are as follows.

#### 2.1.1. 18-Item Experiences in Close Relationships—Revised (ECR-R-18) 

This self-rating tool uses a five-point Likert type to rate attachment anxiety and avoidance in adults [42]. The Thai version has nine questions in each subtype [43]. The avoidance questions are reversed, and the mean totals of the subscales are used. In this study, the questions were adapted by changing “romantic partner” to “parents.” This measure was found to have adequate content validity and internal consistency in children and adolescents in different cultures [44,45]. The Cronbach’s alpha was 0.829 on the anxiety subscale and 0.724 on the avoidance subscale in a study of 466 Thai university students. The tool was previously tested by 40 day-school students with Cronbach’s alpha was 0.845 for attachment anxiety, and 0.844 for attachment avoidance. In the present study, Cronbach’s alpha was 0.864 on the anxiety subscale and 0.837 on the avoidance subscale.

#### 2.1.2. Inner-Strength-Based Inventory (I-SBI)

This questionnaire measures ten positive behavioral characteristics inspired by the Buddhist ten perfections (e.g., meditation, truthfulness, perseverance). Each characteristic has multiple-choice responses along a five-point scale. Mean scores are totaled for each item. The person reliability is 0.86 by Rasch analysis. A two-week test-retest by intraclass coefficient was 0.88 [46]. Meditation frequency was determined by selecting an item from this inventory. All five ratings are “I rarely meditate, or I have never properly meditated before” (1 point), “I try to meditate on some occasions” (2 points), “I often meditate but not every day” (3 points), “I meditate every day, at a certain time” (4 points), and “I meditate every day, at a certain time including some other time available” (5 points).

#### 2.1.3. Precept Practice Questionnaire (PPQ) 

This is a six-item self-rating tool with five items that measure five-precept practice in Buddhism [28], i.e., refraining from taking life (I avoid killing living things (including animals and insects)), stealing (I avoid taking someone’s belongings without permission), sexual misconduct (I avoid sexual misconduct), false speech (I avoid telling lies), and consuming intoxicants (I avoid alcohol drinking and substance use). The precept questions measure the frequency with which a person refrains from the behavior using a five-point Likert scale from 1–5 with labels about adherence frequency (never, rarely, sometimes, often, and always). The sixth question assesses a person’s motivation for refraining from the precepts, i.e., amotivation (I have no idea why I avoid these behaviors), intrinsic (avoiding these behaviors is good for everyone), and extrinsic (I want others to see me as a good person). The questionnaire was previously tested by 40 day-school students with good reliability. The Cronbach’s alpha amongst the present boarding school student sample was 0.850. See a copy of the English questionnaire in Appendix A.

#### 2.1.4. Resilient Inventory (RI-9)

The scale measures a person’s ability to recover from setbacks and problems. It has nine items using a five-point Likert type of scale with choices from “1—does not describe me at all,” to “5—it describes me very well.” The total scores range from 9–45, with higher scores indicating more resilience. The person reliability is 0.86 using Rasch analysis and the Cronbach’s alpha value is 0.90 [47]. Cronbach’s alpha amongst this study’s sample was 0.86.

All significantly correlated variables were tested in the hypothesis model to find the mediation effects amongst the variables and to see how precept or meditation practices influence the relationship between attachment and resilience. An indirect effect of attachment on resilience through meditation practice and precepts was tested.

For mediation analysis, the researchers used the methods discussed by Hayes [48] to examine the relationship between attachment (X) and resilience (Y) through meditation (M1). Furthermore, we tested the model when precepts were included as the second mediator (M2) in a multiple parallel mediation model (Figure 1).

A mediation model within a structural equation model (SEM) framework was applied to examine how well the proposed model fits the data. Using SEM allowed both parallel mediation models of attachment anxiety and attachment avoidance to be tested simultaneously. To test for model fitness, the following fit indices were used: the comparative-fit index (CFI) > 0.95; the Tucker–Lewis Index (TLI) > 0.95; the root mean square error of approximation (RMSEA) < 0.06; and the ratio χ^2^/DF should be <3. [49] The model was tested using a maximum likelihood estimation method for covariance matrices.

The 5000 bootstrap resampling and the product of coefficients as suggested when conducting mediation analysis were performed [50]. Standardized regression coefficients and p-values were reported for the direct effect coefficients and bootstrap confidence intervals for conditional indirect effects and for conditional indirect effects pairwise contrasts. Confidence intervals that did not straddle zero were indicative of statistical significance. For all the analyses, the level of significance was set at *p* < 0.05. All statistical analyses were performed using the IBM Statistical Package for the Social Sciences (SPSS 22) and Amos, version 18 (IBM Corp., Armonk, NY, USA).

## 3. Results

The sample consisted of 453 participants, mostly females and Buddhists, aged between 15 and 18 years old. Slightly over half of the participants were from Buddhist boarding schools. Details of the scores of each measurement are shown in Table 1.

The dispersion and distribution of the measurement scores were described in Table 2. According to the mean score of I-SBI: Meditation, this population meditated almost every day. Precepts were adhered to often.

Zero-order correlations between variables are shown in Table 3. Being female was significantly and highly related with meditation and observing precepts (*p* < 0.01), whereas age did not significantly correlate with attachment, resilience, meditation, and precept practice. As expected, Buddhist identity was positively associated with meditation (*p*< 0.01), as were Buddhist schools (*p* < 0.01). Among sociodemographic variables, family income was also found to be significantly associated with all studied variables (all *p* < 0.01).

Figure 2 shows the direct effects of the mediation model. Both meditation and precepts partially mediated the relationship between attachment and resilience scores. Meditation practice and precept practice mediated the relationship between attachment anxiety and resilience (β = −0.15, *p* < 0.001), and attachment avoidance and resilience (β = −0.11, *p* < 0.05). Notably, a nonsignificant path between attachment avoidance and precept practice was observed (β = −0.09, *p* > 0.05), and attachment anxiety had a stronger effect than attachment avoidance on resilience scores. The total indirect effect of attachment anxiety was β = −0.086, 95%CI = −0.125, −0.054, *p* < 0.001, whereas attachment avoidance was β = −0.050, 95%CI = −0.088, −0.021, *p* = 0.006. The indirect effect of attachment anxiety through meditation was β = −0.047, 95%CI = −0.078, −0.025, *p* < 0.001, whereas attachment avoidance through meditation was β = −0.032, 95%CI = −0.059, −0.010, *p* = 0.017. The indirect effect of attachment anxiety through precepts was β = −0.042, 95%CI = −0.072, −0.021, *p* < 0.001, whereas attachment avoidance through precepts was β = −0.021, 95%CI = −0.051, 0.002, *p* = 0.129. The indirect effect through meditation was significantly higher than precepts. 

Regarding the effect of covariates, age positively predicted RI scores (β = 0.11, *p* = 0.009). Likewise, income negatively predicted precept practice scores (β = 0.11, *p* = 0.018). Secular school negatively predicted both meditation practice and precept practice scores (β = −0.32, *p* < 0.001, and β = −0.12, *p* = 0.018, respectively). Religion (Buddhism) was positively associated with meditation (β = 0.18, *p* < 0.001). The best fit model resulted in the following fit statistics; CFI = 1.000, TFI = 1.001, RMSEA = 0.000, SRMR = 0.005, chi-square = 0.990, df = 1, *p* = 0.320). This model can explain the 33% variance of the RI scores, compared with 23% from the direct effect of attachment anxiety and avoidance.

## 4. Discussion

This study aimed to explore the relationship between attachment and resilience, and how meditation and precept practices influence this relationship among boarding school adolescents. The significant negative association between attachment anxiety and avoidance and resilience highlights that insecure attachment corresponds with low levels of resilience, consistent with many related studies [15,51]. The fact that both attachment anxiety and avoidance negatively predicted the resilience score suggests that high levels of secure attachment correspond to a high level of resilience, while high levels of fearful attachment correspond to a low level of resilience [52].

Meditation and precept practices partially mediated the relationship between attachment insecurity and resilience. In other words, the magnitude of the effect of insecure attachment on resilience was reduced by meditation practice and precept adherence. Nevertheless, the explanation of the total effect of attachment on resilience increased to 33.3% when the indirect effect of attachment through meditation and precept adherence was applied. Mediators of attachment and mindfulness traits have included thought suppression, rumination, and attentional control [53], and associations were found between mindfulness traits and high levels of adaptive emotion regulation and low levels of psychopathology and negative emotion [30], factors relevant to meditation practice within this research that may explain the model increase. The findings are in line with related research that emotional regulation was related to resilience amongst Thai adolescents in Bangkok [54]. Precept and meditation practices are considered inner strengths [46] which are readily available and which, with practice, can enhance a person’s resilience in adolescence. As resilience is a process of adaptation to adverse experiences, practicing meditation and refraining from breaking precepts may support resilient growth among this population.

Notably, there was a nonsignificant path between attachment avoidance and precept practice (β = −0.09, *p* > 0.05). The larger mediation effect size of attachment anxiety compared with attachment avoidance is partly explained by the non-significant path between attachment avoidance and precept practice, implying that attachment-anxious people are less likely to follow the precepts. The dominant role of attachment anxiety among this adolescent sample is supported by related studies on older populations. For example, anxious attachment was found to predict suicidality in elderly Thai populations [8,9]. Moreover, attachment anxiety predicted loneliness and depression among long-term care residents [55].

Precepts have had limited research and are not well described in the psychological literature. Refraining from those behaviors can be viewed as self-regulation if related to attachment. Research showed that fearful insecurity was associated with maladaptive behavior concerning sexual misconduct, one of the five precepts [56]. We hypothesize that precepts are not mere behaviors that require self-control; they also result in wisdom that cultivates a person’s generosity, respect, honesty, mindfulness, and responsibility for others [46]. This could explain why precepts are related to both attachment anxiety and attachment avoidance.

In line with other related research, age is associated with resilience [57] as it involves a developmentally cognitive maturation in advancing age. By contrast, income is negatively associated with precept practice. Thailand is an agricultural country; most of the parents who have a lower level of family income work in rice fields or a farm, and the parents might not expect children to strictly observe the precept against killing (e.g., insects or farm animals killed for food). Consistent with the previous study, financial security concerns were found to predict noncompliance with the precepts for people in lower-income households [58]. High-income families are likely to have more opportunities to teach morals and precept adherence to their children.

As expected, Buddhist schools were associated with higher levels of meditation and precept practices. The Buddhist schools have a period every school day during which students can meditate. Some students meditate more than the single mandatory meditation period. However, after controlling for those related factors, attachment anxiety is still mediated by meditation and precept practices, whereas attachment avoidance is mediated only by meditation practice.

### 4.1. Research Implication 

Our study has both clinical and research implications. Both attachment insecurity and resilience are important for adolescents coping with mental health problems. For example, resilience was also negatively associated with externalizing difficulties, and interventions for adolescents have been shown to be effective in reducing anxious and depressive symptoms [59]. A previous study found that mindfulness meditation partially mediated the relationship between insecure attachment and depression, anxiety, and stress amongst adolescents [60]; the present study sheds some light on the importance of observing the five precepts along with the effect of meditation. 

This research confirms and extends earlier attachment research examining the relationship between attachment and mental health; in addition, it enhances our understanding of the benefits of Buddhist religious practices in Thailand on mental health. This research on boarding schools in Thailand highlights the positive mental health associations of meditation and precept adherence on adolescent resilience. This study may inform schools, policy makers, future intervention studies, and further research about the ways in which meditation and moral precept practice in schools influences mental health outcomes. 

### 4.2. Limitations

Our findings should be generalized with caution due to the participants’ status as boarding school students. Two of the boarding schools were Buddhist; they therefore supported and facilitated religious values and practices, and all five participant boarding schools provided more homogeneous living environments than day schools. However, parity has been found between attachment and mental health outcomes amongst day school and boarding school students [41]. Another concern is that the cross-sectional design may preclude a definite conclusion of causal relationship. Longitudinal or experimental designs will be needed to fully explicate these complex relationships.

## 5. Conclusions

This study provides evidence that meditation and precept practices mediate the relationship between insecure attachment and resilience. Secure attachment was positively associated with precepts, meditation, and resilience, and insecure attachment was negatively associated with all three factors. The participants meditated almost every day, and generally closely adhered to the Buddhist precepts. This research may help explain how adolescents enhance their resilience despite experiencing insecure attachment. Future research should examine the combination of practicing precepts and meditation. Studying the influence of precept adherence in non-Buddhist culture also deserves consideration in future research. 

## Figures and Tables

**Figure 1 children-09-00371-f001:**
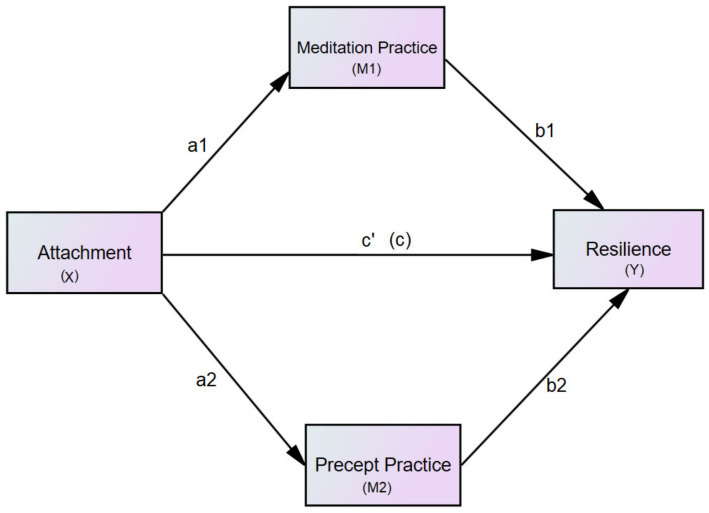
The hypothesized model. Notes: Path diagram illustrating the direct effects and causal paths linking attachment and resilience. a1, a2, b1, b2, c, and c’ = path coefficient (standardized coefficient), x = antecedent variable, M1 = mediator 1, M2 = mediator 2, Y = outcome.

**Figure 2 children-09-00371-f002:**
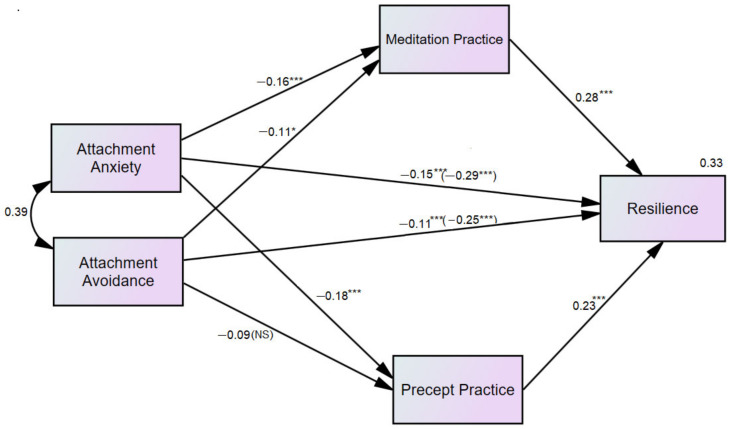
Parallel mediation model (*n* = 453). Indirect effects of attachment anxiety/attachment avoidance on resilience through meditation and precept practice (controlling for covariates). The model is controlled for age, sex, school type, religion, and family income. Standardized effects are presented. The effects on the direct path from attachment to resilience depict the direct effect and the (total effect) * *p* < 0.05, *** *p* < 0.001, NS = non-significant.

**Table 1 children-09-00371-t001:** Descriptive statistics of sample characteristics.

Variables (*n* = 453)	*n* (%) or Mean (SD)
Sex, female	398 (87.9)
Age (years)	16.35 ± 0.96
Religion	
Buddhist	404 (89.2)
Christianity	49 (10.8)
Family income—less than USD 295 *	244 (53.9)
Family income—USD 296 and higher	209 (46.1)
School types—Buddhist	242 (53.4)
School types—Secular	211 (46.6)
Meditation Type	
Breathing meditation	129 (28.5)
Buddha image kasina	115 (25.4)
Manomayiddhi	85 (18.8)
Mindfulness occupied with the body	47 (10.4)
Death contemplation	46 (10.2)

* 1 USD = 32 THB; SD = Standard deviation; Manomayiddhi is a combination of several meditation types, e.g., breathing meditation, Buddha image kasina, death contemplation, etc.

**Table 2 children-09-00371-t002:** Descriptive characteristics of the measurements (*n* = 453).

Measurement	Mean ± SD	Skew	Kurtosis
ECRR-avoidance	2.80 ± 1.2	0.84	0.54
ECRR-anxiety	2.76 ± 1.1	0.44	−0.43
PPQ	4.00 ± 0.9	−1.30	1.17
I-SBI-Meditation	2.94 ± 1.4	1.59	−1.30
RI-9	33.77 ± 5.8	−0.94	−0.624

ECRR-anxiety = the experience of close relationship questionnaire (revised), anxiety dimension; ECRR-avoidance = the experience of close relationship questionnaire (revised), avoidance dimension; PPQ = Precept Practice Questionnaire; I-SBI = Inner-Strength Based Inventory: Meditation; RI-9 = the 9-item Resilience Inventory; SD = Standard deviation.

**Table 3 children-09-00371-t003:** Correlation between variables (*n* = 453).

	1	2	3	4	5	6	7	8	9	10
1. Sex	−	−0.04	−0.06	−0.22 **	0.19 **	0.10 *	0.04	−0.08	−0.14 **	−0.12 **
2. Age		−	−0.10 *	−0.15 **	0.12 **	−0.06	−0.06	0.08	−0.01	−0.02
3. Religion			−	0.19 **	−0.04	−0.02	−0.11 *	0.13 **	0.22 **	0.06
4. Income				−	−0.25 **	−0.12 **	−0.12 **	0.18 **	0.19 **	0.19 **
5. School Type					−	0.30 **	0.15 **	−0.28 **	−0.39 **	−0.21 **
6. ECRR_Avoidance						−	0.38 **	−0.32 **	−0.28 **	−0.21 **
7. ECRR_Anxiety							−	−0.35 **	−0.28 **	−0.24 **
8. RI								−	0.43 **	0.36 **
9. Meditation									−	0.19 **
10. PPQ										−

ECRR-anxiety = the experience of close relationship questionnaire (revised), anxiety dimension; ECRR-avoidance = the experience of close relationship questionnaire (revised), avoidance dimension; PPQ = Precept Practice Questionnaire; RI-9 = the 9-item Resilience Inventory. * *p* < 0.05, ** *p* < 0.01.

## Data Availability

The datasets used and/or analyzed during the current study are available from the corresponding author upon reasonable request.

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
