# Peer review of "Meditation and Five Precepts Mediate the Relationship between Attachment and Resilience"

_children, 2022, doi:10.3390/children9030371_

Round 1

Reviewer 1 Report

The connection between attachment and resilience is of general interest, as are the role of mindfulness and meditation. However, the manuscript shows substantial flaws. To specifiy:

  1. The introduction and theoretical background are incoherent in parts, constructs are presented but not connected with each other. Some statements are simply wrong or overly generalizing, such as “Secure attachment is viewed as a norm and insecure attachment as a pathology” or “Caregiver- infant attachment is stable throughout life and relationships”.
  2. Mindfulness is not explicitly part of the research presented here, however it is touched upon and not connected with other theoretical background. Meditative/mindfulness attributes are mentioned but not defined, and so on.
  3. Mechanisms via which attachment influences mindfulness/meditation are not described, whereas the precepts refer to the absence of some aspects that could be termed “problem behavior” but are not theoretically linked to the literature on this and on self regulation.
  4. I am concerned that the ECR, a tool designed to measure attachment in adult relationships, may not be appropriate in 16 year old Thai boarding school female adolescents as can be seen in the very low SD. A similar concern regards the PPQ-questionnaire. One cannot help but wondering if the emerging results can actually be interpreted. This would have to be argued in more detail.
  5. Finally, the clearly not very objective evaluation of this religious path, especially in line 250-257 does not appear to be informed by scientific neutrality. What is to be understood from “cognitive and emotional virtues“? And (line 259) attachment anxiety and attachment avoidance are in no way characterized by being the emotional and the behavior part. Both attachment orientations influence emotions, cognitions and behaviors. This is such a basic error that I do not feel comfortable recommending this paper for publication.

Author Response

Psychotherapy/Personality disorder Clinic and Education Center, 

Psychotherapy Unit & Geriatric Psychiatry Unit, Department of Psychiatry, 

Faculty of Medicine, Chiang Mai University, Chiang Mai,

Kingdom of Thailand. 50200

Dear Editor

Re: Reviewers’ comments on manuscript ID 1522782, Meditation and Five Precepts Mediate the Relationship Between Attachment and Resilience, dated 11 Feb 2022

            Thank you for your consideration in publishing our article. We are thankful to reviewers for their useful comments. Please see below for our point-by-point responses to the reviewers’ comments. The revised ones are in green color. The manuscript is also edited again for more clarification.

Reviewer #1

General Comment – The connection between attachment and resilience is of general interest, as are the role of mindfulness and meditation. However, the manuscript shows substantial flaws.

Author Response: This publication did, indeed, require significant rewriting for which the primary author is grateful to you for highlighting. We have corrected what were mistakes within the theoretical background.

Comment #1 - The introduction and theoretical background are incoherent in parts; constructs are presented but not connected with each other. Some statements are simply wrong or overly generalizing, such as “Secure attachment is viewed as a norm and insecure attachment as a pathology” or “Caregiver- infant attachment is stable throughout life and relationships”.

Authors Response: Thank you very much. We now have the Introduction rewritten. Regarding the statements that you raised here; we have them revised. It now reads ‘Secure attachment is viewed as a healthy attachment norm and insecure attachment as a psychopathology [2]. Caregiver-infant attachment is mostly stable throughout life and relationships, having an enduring influence on future attachment bonds [4,5] and significant influence on behavior [6] and mental health problems [7-11].’ Introduction, first paragraph.

Comment #2 - Mindfulness is not explicitly part of the research presented here, however it is touched upon and not connected with other theoretical background. Meditative/mindfulness attributes are mentioned but not defined, and so on.

Authors Response: Thank you for bringing up this issue. Indeed, the use of mindfulness was not clear previously. Mindfulness is not a study variable, but supportive evidence ties together the association between attachment and mindfulness. We have rewritten the introduction section about meditation to clarify its use, having defined and connected mindfulness in relation to attachment and resilience, and we have clarified mindfulness meditation as opposed to mindfulness traits (awareness, receptiveness). These uses have bearing on this research’s study of meditation and possible links between attachment, meditation, and mental health outcomes.

According to the Tripitaka (Buddhakosa, 2010), mindfulness is a stage of recollection. It is part of meditation. There are forty kinds of samadha meditation and 9 vipassana meditations (insight or knowledge meditation). Samadha meditation includes ten recollections (or mindfulness), ten kasina (e.g., colour), ten kinds of foulness, four meditations on the elements (i.e., earth, water, air and fire), four meditations on formless fear, meditation on sublime stage of mind, and meditation on repulsiveness in nutriments. Ten kinds of recollection are recollection of the Buddha, recollection of the Dhamma, recollection of the Sangha (noble monks), recollection of virtue, recollection of generosity, recollection of deities, recollection (or mindfulness) of death, mindfulness occupied with the body, mindfulness of breathing, and recollection of peace. Examples of vipassana (or insight) meditations are impermanence, state of sufferings, soullessness, etc. ‘Breathing meditation’ and ‘body scan meditation’ are the most often practiced. In the current research all kinds of meditation performed among the participants’ culture are included. Eight common kinds were recorded in the questionnaire. The current research, however, has highlighted mindfulness meditation as research is available about this practice.

We revised the section on meditation by adding the text ‘Meditation includes a range of methods that are categorized into two types i.e., samatha (e.g., breathing meditation, mindfulness occupied with the body, recollection of dhamma/Buddhist teaching), vipassana or insight meditation (e.g., impermanence).’ Paragraph 3 has been extensively revised to clarify the relationship between meditation and mindfulness. Moreover, in the material and methods section, we have added the text All meditative styles practiced were accepted’ in the first paragraph. For iSBI, we deleted the word ‘mindfulness’ to prevent confusion. Results of common types of meditation were also added in Table 1.

Comment #3: Mechanisms via which attachment influences mindfulness/meditation are not described, whereas the precepts refer to the absence of some aspects that could be termed “problem behavior” but are not theoretically linked to the literature on this and on self-regulation.

Authors Response: Thank you. We have revised these parts,

For meditation and attachment, it now reads,

Regarding meditation practice, attachment security priming increased the likelihood that new meditators continued to meditate while also increasing distress regulation[31]. Meditation frequency was also associated with higher emotional intelligence, self-efficacy, and lower perceived stress amongst Thai people[32]. Attachment security has been shown to be associated with mindfulness (awareness, receptiveness), which theoretically forms a coherence of mind that impacts psychological and physical health, emotional regulation, and relationship quality [33]

Attachment anxiety and avoidance were strongly and negatively associated with mindfulness [34,35]

For precepts and attachment, it now reads,

Intrinsic religious orientation, which includes using religious teachings to guide behavior, has been shown to be a protective factor in adolescents against early onset alcohol use[22] and teenage delinquency[23]. These risk behaviors (non-adherence to precepts) cluster amongst Thai adolescents [24,25], and individual insecure attachment styles that are associated with lower affect regulation, higher impulsive behaviors [26], and higher incidents of substance use [27], and they increase other behavioral risks[25].

Comment #4: I am concerned that the ECR, a tool designed to measure attachment in adult relationships, may not be appropriate in 16 year old Thai boarding school female adolescents as can be seen in the very low SD. A similar concern regards the PPQ-questionnaire. One cannot help but wondering if the emerging results can actually be interpreted. This would have to be argued in more detail.

Authors Response:  Thank you for asking this important question. We understand that the ECR-R is intended to measure adult relationships. However, based on the suggestion of Chris Fraley, the creator of the ECR-R, the ECR-R can be adapted to other significant relationships (R. Chris Fraley, URL: http://labs.psychology.illinois.edu/~rcfraley/measures/ecrr.htm). More importantly, the ECR-R has previously been used in adolescent populations (Yang, 2008), with a female majority (Fernandez-Fuertes, 2011) from different cultures, and it has been shown to have adequate content validity and internal consistency. Before we started the data collection, the tool was tested with a population of 40 day school students, and it was shown to have a Cronbach’s alpha of 0.845 for attachment anxiety, and 0.844 for attachment avoidance. We revised the information about the tool in the materials and methods section. Please see ECR-R-18.

Regarding the standard deviation (SD), in general, there is no “acceptable” or “unacceptable” SD, but this statistic provides information about the distribution of data. For ECR-R, we found that the distribution is consistent with what have been reported in many studies. Please see below for reliability and ECRR mean item score and SD comparisons with the above cited research.

Reliability

The current research ECRR reliability statistics were:  anxiety (α = 0.84) and avoidance (α = .86). Yang, (2008) reported similar ECRR reliability - anxiety (α = .75) and avoidance (α = .77).  Likewise, Fernandez-Fuertes (2011) reported similar data:  ECRR reliability - anxiety (α =.83) and avoidance (α =.86)

Mean scores and SD

The current research: (anxiety 2.76 ± 1.1, avoid 2.80 ± 1.2)

Fernandez-Fuertes, (2011): (anxiety 3.92 ± 0.94, avoid 3.12 ± 1.06)

Yang (2008): (anxiety 3.08 ± 0.80, avoid 3.08 ± 0.84).

Fraley (personal communication): (anxiety 3.56 ± 1.12, avoid 2.92 ± 1.19), based on a sample of 17,000 people.

Likewise, the PPQ has a mean of 4.00, and a SD of 0.9 (range 1-5). This documents that most of the sample practiced five precepts. However, the SD of 0.9 is not considered too narrow compared to the ECR-R that has the range between 1 to 7.  The distribution of the PPQ is close to normal in shape (bell curve). Statistically, 68% of the scores in a normal distribution are plus or minus one standard deviation;  in our sample, 72.8% of the sample has the score within 1SD, suggesting that the PPQ’s distribution is legitimate.

Comments #5: Finally, the clearly not very objective evaluation of this religious path, especially in line 250-257 does not appear to be informed by scientific neutrality. What is to be understood from “cognitive and emotional virtues“? And (line 259) attachment anxiety and attachment avoidance are in no way characterized by being the emotional and the behavior part. Both attachment orientations influence emotions, cognitions and behaviors. This is such a basic error that I do not feel comfortable recommending this paper for publication.

Authors Response: Thank you for your detailed, candid critique. We have deleted those sentences that are not related to scientific neutrality, as suggested, as well as the confusing discussion of cognitive and emotional virtues.  It now reads,

We hypothesize that precepts are not mere behaviors that require self-control, but also wisdom that cultivates a person’s generosity, respect, honesty, mindfulness, and responsibility for others [46]. This could explain why precepts are related to both attachment anxiety and attachment avoidance.

Reviewer #2

General Comment: Thank you for selecting such an interesting topic which I hope to increase the readership of the journal. Yet, there are certain drawbacks that should be taken due care of.

Author Response: Thank you. We do want to make a meaningful contribution to the existing literature. We have taken time to clean up the previous version, and we have addressed each your comments below.

Comment #1 - The literature does not provide readers with a deep and thorough background of the study variables. Current studies should be reviewed to add to justify the

significance of the study.

Author Response: Thank you. We have taken time to support the study variables per your recommendation. Attachment, Precepts, Meditation (and its links with mindfulness), Resilience, and how these variables relate have all been given more attention and clarity.  

Comment #2 – Editing and proofreading is a must.

Author Response: Thank you. We have since taken more time to do both.

Comment #3 - A clear and detailed explanation of the methods of the study should be provided to give readers, scholars and practitioners the opportunity to do similar studies or implement the study.

Author Response: Thank you. We have added more information about methods and instruments to ensure understanding of this research.

Comment #4 – The study conclusions should be elaborated.

Author Response: We have expanded the conclusion with details about meditation and precept adherence. Thank you.

Hopefully, our revision would be sufficient and satisfy the editor and reviewers. We have corrected many other errors and misspellings throughout the manuscript. Thank you for your consideration again. We are looking forward to hearing from you soon.

Best regards,

Prof. Tinakon Wongpakaran, MD, FRCPsychT

Reviewer 2 Report

Dear Authors,

Thank you for selecting such an interesting topic which I hope to increase the readership of the journal. Yet, there are certain drawbacks that should be taken due care of. These are the following:

  1. The literature does not provide readers with a deep and thorough background of the study variables. Current studies should be reviewed to add to justify the significance of the study.
  2. Editing and proofreading is a must.
  3. A clear and detailed explanation of the methods of the study should be provided to give readers, scholars and practitioners the opportunity to do similar studies or implement the study.
  4. The study conclusions should be elaborated. 

Author Response

(The authors gave the same response as above.)

Round 2

Reviewer 1 Report

I feel the manuscript has greatly improved in several important ways. In the introduction, there are, from my point of view, two errors that should be addressed. I must not have pointed that out clearly enough before:

"Secure attachment is viewed as a healthy attachment norm and insecure attachment as psychopathology". -> In my eyes, this is wrong. Insecure attachment is a risk factor for psychopathology, and also found more frequently in clinical populations, but is not in itself a psychopathology.

"Caregiver-infant attachment is mostly stable throughout life and relationships, having an enduring influence on future attachment bonds and significant influence on behavior and mental health." -> To my knowledge, attachment styles show a MEDIUM continuity. Therefore, I do not believe it useful to describe them as "mostly stable". I would go with "has a significant impact" through life or something along these lines.

Other than that, as I said, I do not have further objections to this article being published. For sure, these topics deserve attention.

Author Response

Psychotherapy/Personality disorder Clinic and Education Center, 

Psychotherapy Unit & Geriatric Psychiatry Unit, Department of Psychiatry, 

Faculty of Medicine, Chiang Mai University, Chiang Mai,

Kingdom of Thailand. 50200

Dear Editor

Re: The reviewer’ second comments on manuscript ID 1522782, Meditation and Five Precepts Mediate the Relationship Between Attachment and Resilience, dated 24 Feb 2022

            We are thankful to the reviewer for his/her useful comments. Please see below for our point-by-point responses to the reviewer’ comments. The revised ones are in blue color.

1."Secure attachment is viewed as a healthy attachment norm and insecure attachment as psychopathology". -> In my eyes, this is wrong. Insecure attachment is a risk factor for psychopathology, and also found more frequently in clinical populations, but is not in itself a psychopathology.

Response: Thank you for this detailed comment. We have revised it as suggested.

It now reads, “Insecure attachment is a risk factor for psychopathology, and also found more frequently in clinical populations, but is not in itself a psychopathology.

2."Caregiver-infant attachment is mostly stable throughout life and relationships, having an enduring influence on future attachment bonds and significant influence on behavior and mental health." -> To my knowledge, attachment styles show a MEDIUM continuity. Therefore, I do not believe it useful to describe them as "mostly stable". I would go with "has a significant impact" through life or something along these lines.

Response: Thank you for this helpful comment. We have revised it as suggested. It now reads, “Caregiver-infant attachment has a significant impact through life and relationships, having an enduring influence on future attachment bonds and significant influence on behavior and mental health.

Thank you for your consideration again. We are looking forward to hearing from you soon.

Best regards,

Prof. Tinakon Wongpakaran, MD, FRCPsychT
